# Coping with Tissue Sampling in Suboptimal Conditions: Comparison of Different Tissue Preservation Methods for Histological and Molecular Analysis

**DOI:** 10.3390/ani11030649

**Published:** 2021-03-01

**Authors:** Arturo Nicoletti, Paola Pregel, Laura Starvaggi Cucuzza, Francesca Tiziana Cannizzo, Alessandra Sereno, Frine Eleonora Scaglione

**Affiliations:** Department of Veterinary Sciences, University of Turin, Largo P. Braccini 2, 10095 Grugliasco, Italy; paola.pregel@unito.it (P.P.); laura.starvaggicucuzza@unito.it (L.S.C.); tiziana.cannizzo@unito.it (F.T.C.); alessandra.sereno@unito.it (A.S.); frineeleonora.scaglione@unito.it (F.E.S.)

**Keywords:** formalin, silica beads, tissue preservation, under-vacuum

## Abstract

**Simple Summary:**

Specimen collection and preservation can be challenging, especially when suboptimal conditions occur. A relevant amount of time could be required from sampling to the tissue analysis, and therefore a high-quality conservation technique is vitally important for diagnostic purposes. The aim of this study was the identification of a reliable and economic method for tissue preservation to be used in complex “in-field” situations, suitable for both histological and molecular analysis. Punch biopsies were collected from six cattle livers. Comparison among methods of preservation using RNAlater, silica beads, or under-vacuum was carried out using different times and temperatures. Three days were assumed as a maximum time interval from sampling to laboratory and 4 °C and 24 °C chosen as references for refrigeration temperature and room temperature, respectively. Histological and biomolecular analyses were performed. RNAlater and silica beads poorly preserved the histological parameters evaluated; conversely, vacuum-sealed samples showed a good grade of preservation for 48 h. DNA quality was acceptable for each sample. RNA integrity showed promising results for samples preserved with silica beads.

**Abstract:**

A high quality of samples is crucial for the success of the analysis and diagnostic purposes, and therefore the right method of conservation is vitally important for an optimal preservation of tissues. Indeed, the time to deliver the sample to the laboratory could be remarkably long, especially under suboptimal conditions, and the use of specific fixatives or cold storage may not be possible. Moreover, the portability and cost of storage equipment, their toxicity, and their ease of use play a central role when choosing the correct preservation method. The aim of this study was the identification of a reliable and economic method for tissue preservation, to be used in “in-field” sampling, suitable for both histological and molecular analysis. Punch biopsies were collected from six cattle livers. Comparisons among methods of preservation using RNAlater, silica beads, and under-vacuum was carried out. These methods were tested through considering different times and temperatures, assuming three days as a maximum time interval from sampling to laboratory and choosing 4 °C and 24 °C as references for refrigeration temperature and room temperature, respectively. Histologically, the integrity of nucleus, cytoplasm, preservation of liver structure, and easiness of recognition of inflammatory infiltrate were evaluated. The integrity of the extracted DNA and RNA was evaluated through PCR and by means of an automated electrophoresis station, respectively. RNAlater and silica beads poorly preserved the histological parameters evaluated, independently from the temperature. Conversely, the vacuum-sealed samples showed a good grade of preservation until 48 h. DNA quality was acceptable for each sample. RNA integrity showed promising results only for samples preserved with silica beads.

## 1. Introduction

Samples’ conservation, from collection to laboratory, is critical for the success of the analyses and diagnostic purposes. Sometimes, when cold storage may not be possible, for instance, sample preservation can be challenging. This is particularly true in forensic science and in “in-field” critical situations, such as mass disasters, when successful DNA sampling is fundamental for the victim’s identification. Multiple factors should be analyzed to reach a full understanding of this issue, especially when evaluating tissue preservation protocols in sub-optimal conditions. In fact, not only the efficiency of the method but also toxicity, portability of storage equipment or materials, cost, and ease of use play a central role in guiding the correct choice [1].

Fixatives are fundamental in order to obtain reliable slides for histological analyses. Formalin, for example, is a chemical extensively used in anatomy and pathology, and it is the most widely used fixative for tissue preservation and morphology analysis [2]. It is characterized by its cheapness, easy storage, ease of use, and its guarantee of long-lasting preservation of cells’ morphology and tissue architecture [3]. However, two non-negligible side effects should be taken into account: toxicity and poor-quality biomolecular conservation. Formaldehyde has irritating effects and cancerogenic properties, and indeed it was catalogued as a primary carcinogenic substance by the International Agency for Research on Cancer (IARC) [4]. It also seems to be associated with an increased risk of birth malformations, spontaneous abortions, immune disease, neurotoxicity, and cognitive dysfunction [5,6,7]. Molecular analyses are poor in quality when performed on formalin fixed tissue because of DNA fragmentation and chemical modification, especially during long-term storage [8,9,10]. Formalin fixation is also considered a limiting factor for immunohistochemical analysis because formalin can cross-link antigens and mask epitopes [11,12,13].

Regarding nucleic acid conservation, liquid nitrogen is considered the gold standard for tissue storage in terms of guaranteeing an optimal biomolecular quality for future analysis [14,15]. As an alternative, ultralow temperature freezer can be used, but usually both preservation methods are not available immediately after tissue collection [16], especially in “in-field” conditions. In this framework, RNAlater can be a valid solution for molecular analysis, especially on-site, although it is very expensive compared to other methods [17]. Despite these considerations, there are some discrepancies regarding the impact of RNAlater on the preservation of tissue morphology. In some studies, RNAlater maintained a satisfactory tissue morphology [16], while in others, tissue morphology was poorly preserved [10] or substantial differences were recorded among different tissues [18].

In order to find a reliable method for both histological and biomolecular analyses, researchers have taken other techniques into account. Under-vacuum storage was considered as an alternative to formalin to overcome the banning of formalin because of its toxicity [19]. In fact, it can also be used as a reliable method to transport samples from surgery to pathology laboratories, at 4 °C, blocking enzyme activity and preventing tissue autolysis [20,21]. To the best of our knowledge, the use of under-vacuum methods for concurrent histologic and biomolecular preservation in the biomedical field has not been explored yet.

Another well-known and widely used procedure to preserve samples is to block enzymatic activities (especially nucleases) through quick and complete desiccation, since water is fundamental for enzymatic activity [17]. Silica beads are a recognized desiccant, extensively used in many fields. In biomolecular field, they were tested for long-term preservation of brown bear DNA fecal samples [22]; DNA preservation in bat specimens, with excellent results [23]; and as a desiccation mean for amniotic membrane sampling for ocular surface reconstruction [24]. Nowadays, no research evaluating tissue preservation for both histopathological and biomolecular analysis in the biomedical field is available.

Regardless of the method taken into account, the temperature is a fundamental parameter to be considered. In fact, enzymatic activity is widely influenced by temperature, and therefore autolysis and putrefaction are strictly dependent on temperature [25,26,27,28]. More accurately, temperature and rate of decomposition can be correlated by Van’t Hoff’s Law [29]. The aim of our study was the identification of an efficient, non-toxic, economic, easily portable, and easy-to-use method for tissue preservation, reliable for both histological and molecular analysis and suitable for “in-field” sub-optimal conditions, forensic medicine, and veterinary toxicology, being capable of improving the performance of tissue sampling in challenging conditions.

## 2. Materials and Methods

### 2.1. Sampling

A portion of six cattle livers (*Bos taurus*, Linnaeus, 1758), without macroscopically evident lesions, was collected within 1 h after the slaughter at the Department of Veterinary Sciences’ slaughterhouse (University of Turin). Forty-two samples were collected by means of a punch (diameter 6 mm, length 2 cm).

Samples were processed as follows (Figure 1):-One sample (T0) was collected and immediately fixed in 10% neutral buffered formalin and used as a control for histological analysis. Another punch biopsy (T0) was immediately put into RNAlater solution and stored for 72 h at 4 °C; then, the solution was discarded, and the sample was immediately frozen at −80 °C and used as a control for biomolecular analysis.-Ten samples were put into 10 different microfuge tubes with 1.3 mL of RNAlater each.-Ten samples were put into 10 different 10 mL tubes filled with about 5.6 g of silica beads.-Ten samples were vacuum-sealed with double sealing.-Ten samples were put into 10 different 50 mL tubes without preservation treatments.

Half of the samples collected with each method were stored at 4 °C, half at 24 °C.

At fixed times (4, 10, 24, 48, and 72 h), 1 sample from each of the 4 groups preserved at both temperatures was divided into 2 parts: 1 was formalin-fixed and the second part was stored into RNAlater at 4 °C. After further 48 h, each formalin-fixed sample was paraffin-embedded, whereas RNAlater was discarded from preserved samples and samples were cut in portions of around 50–100 mg each and stored at −80 °C for biomolecular analyses.

Sampling was repeated on six different livers with the same methodology.

### 2.2. Histology

Formalin-fixed specimens were sectioned into 3–4 µm tissue sections, hematoxylin and eosin (H&E) stained, and histologically examined. Hepatocytes and inflammatory infiltrate, if present, were assessed. In order to standardize the evaluation of microscopic slides, we histologically considered 4 different parameters (Table 1): (a) morphology, positions, and integrity of nuclei; (b) quality of cytoplasm and presence of hydropic degeneration; (c) preservation of microscopic liver structure and morphology; (d) morphology, integrity, and easiness of recognition of inflammatory infiltrate. Each parameter was graded according to a semi-quantitative scale. The rating categories were (1) completely inadequate, (2) poor, (3) satisfactory, (4) good, and (5) excellent, in comparison to the T0 sample.

### 2.3. Biomolecular Analyses

50–100 mg RNAlater-preserved tissues stored at −80 °C were analyzed. DNA and RNA extractions were performed by means of TRIzol Reagent (Invitrogen, ThermoFisher Scientific, Waltham, MA, USA), according to the manufacturers’ instructions. Briefly, tissues were homogenized through a homogenizer (TissueLyser 2, Quiagen, Hilden, Germany), adding 1 mL of TRIzol Reagent for each sample. After a short incubation (5 min), 0.2 mL of chloroform per 1 mL of TRIzol Reagent was added, and the mixture was incubated for 3 min. Samples were centrifuged for 15 min at 12,000× *g* at 4 °C. The aqueous phase contained the RNA, whereas the interphase contained the DNA. A portion of the RNAlater T0 sample was used as a positive control for the procedures.

#### 2.3.1. DNA Isolation

As reported in the TRIzol Reagent (DNA isolation) User Guide, a precipitation step was performed by adding 0.3 mL of 100% ethanol, and tubes were centrifuged for 5 min at 2000× *g* at 4 °C. DNA pellet obtained was resuspended in 1 mL of 0.1 M sodium citrate in 10% ethanol, pH 8.5, incubated for 30 min, centrifuged for 5 min at 2000× *g* at 4 °C twice, and opportunely resuspended.

#### 2.3.2. RNA Isolation

In accordance with the TRIzol User Guide, we isolated RNA by adding 0.5 mL of isopropanol to the aqueous phase, and then incubating the mixture for 10 min at 4 °C and centrifuging it for 10 min at 12,000× *g* at 4 °C. The supernatant obtained was discarded and RNA pellet obtained was washed in 1 mL of 75% ethanol. The obtained sample was centrifuged for 5 min at 7500× *g* at 4 °C and the pellet was resuspended in 20–50 µL of RNase-free water.

#### 2.3.3. Nucleic Acid Quantification and Analysis

DNA and RNA were spectrophotometrically quantified (BioPhotomer plus, Eppendorf) following a dilution with 5 mM TRIS/HCl (pH 8.5)—1:50 for DNA and 1:70 for RNA samples. In particular, 260/280 nm absorbance ratio was determined to check protein contamination, 260/230 nm absorbance ratio to verify the nucleic acid purity, and 340 nm absorbance to test the nicotinamide adenine dinucleotide (NADH) concentration and, ultimately, the concentration of the nucleic acid sample was then measured (µg/mL).

The integrity of the extracted DNA was evaluated by PCR through the amplification of the GAPDH gene (primers: forward 5’-3’: ATGAGATCAAGAAGGTGGTG; reverse 5’-3’: CCAAATTCATTGTCGTACCAG) with the T100 Thermal Cycler (Bio-Rad, Hercules, CA). The amplifications were performed using 10 µL of a master mix (REDTaq ReadyMix, Sigma-Aldrich, St. Louis, MO, USA), 600 nM of each primer, and 1 µL of DNA template. PCR conditions were heating to 95 °C for 30 s, 60 °C for 30 s, and 72 °C for 30 s, for 32 cycles, followed by a final extension step at 72 °C for 10 min.

Then, an electrophoresis on 2% agarose gel was performed, and the expected band at 190 bp was detected by means of a UV transilluminator (GelDoc 1000, Bio-Rad, Hercules, CA, USA), using the GelPilot (Qiagen, Hilden, Germany) marker. The integrity and quality of the extracted RNA were evaluated by an Experion RNA kit (StdSens Analysis kit, Bio-Rad, Hercules, CA, USA), following the manufacturer’s instructions.

### 2.4. Statistical Analysis

Data were analyzed by means of GraphPad InStat (version 6.00, GraphPad Software, La Jolla, CA, USA). Statistical differences were determined by a two-tailed ANOVA for repeated measures, followed by Tukey’s posttests to assay the time effect, and Dunnet’s posttests to evaluate the influence of the conservation method. The differences were considered statistically significant at *p* < 0.05.

## 3. Results

### 3.1. Histology

All the samples stored at 4 °C showed statistically significant differences with the control samples. RNAlater and silica beads methods mostly tended to worsen the histological parameters evaluated, causing a marked vacuolization of the hepatocytes. Conversely, the vacuum-sealed samples and the controls showed a good grade of preservation until 48 h (Figure 2). The specific results are listed below.

#### 3.1.1. Results for Samples Stored at 4 °C

The nuclei of the samples stored in silica beads and RNAlater showed a statistically significant worsening of the scores within the same method of conservation over time—more precisely, between T4 and T24 (*p* < 0.05), T48 (*p* < 0.01), or T72 (*p* < 0.01), and between T10 and T48 or T72 (*p* < 0.01). Moreover, evaluating the different methods of conservations at the same time, we found a statistically significant worsening (*p* < 0.01) for samples in RNAlater at T10, T24, T48, and T72 and in silica beads at T24, T48, and T72 compared to controls.

The cytoplasm score in silica bead-stored samples showed a statistically significant worsening over time only between T4 and T72 (*p* < 0.05). Moreover, evaluating the different methods of conservation at the same time, we found a statistically significant worsening among the samples in RNAlater for T10, T24, and T48 (*p* < 0.01) and T72 (*p*< 0.05) and silica beads (*p* < 0.01) at any time, in comparison to controls.

The preservation of microscopic liver structure and morphology in the silica bead-stored samples revealed a statistically significant worsening of the scores over time—more precisely, between T4 and T24, T72 (*p* < 0.05), or T48 (*p* < 0.01), and between T10 and T48 (*p* < 0.01). All the samples in silica beads and RNAlater showed a statistically significant worsening when compared with the control samples at any time.

The inflammatory infiltrate of the samples stored in silica beads had a statistically significant worsening of the scores over time—more precisely, between T4 and T48 (*p* < 0.01) or T72 (*p* < 0.05), and between T10 and T48 (*p* < 0.05). All the samples in silica beads and RNAlater showed a statistically significant worsening when compared with the control samples at any time.

#### 3.1.2. Results for Samples Stored at 24 °C

The nuclei of the samples stored in silica beads at 24 °C showed a statistically significant worsening of the scores over time, precisely between T4 and T24 (*p* < 0.01). The vacuum-sealed methods produced a statistically significant worsening effect over time, showing differences between T72 and the other times (*p* < 0.01). The samples without any method of conservations worsened significantly at T72 (*p* < 0.01) compared to any other time. All the samples in silica beads and RNAlater deteriorated significantly when compared to the relative control sample at any time.

The cytoplasm of the samples stored at 24 °C in silica beads showed a statistically significant worsening of the scores over time—more precisely, between T4 and T48 or T72 (*p* < 0.05). Moreover, the samples stored in the vacuum-sealed package revealed a significant deterioration between T72 and each one of the other times (*p* < 0.01). Lastly, also for the samples without any method of conservations, we found significant differences between T72 and T4, T10, or T24 (*p* < 0.05). Considering the different methods of conservations at the same time, we found a statistically significant worsening (*p* < 0.01) of the scores for the samples in RNAlater for T4, T10, T24, and T48 and in the samples stored with silica beads at any time, in comparison to control samples.

The microscopic liver structure and morphology of the samples stored in RNAlater had a statistically significant worsening of the scores over time, more precisely between T4 and T72 (*p* < 0.05). In addition, the samples stored in the vacuum-sealed package showed significant differences between T72 and each one of the other times (*p* < 0.01). Considering the different methods of conservations at the same time, we found a statistically significant worsening of scores between the samples in RNAlater for T4 (*p* < 0.05), T10, T24, T48 (*p* < 0.01), and silica beads at any time, in comparison with control samples.

The inflammatory infiltrate of the samples stored in the vacuum-sealed package had statistically significant worsening between T72 and T4, T10, or T24 (*p* < 0.01), and the samples without any method of conservation had significant differences between T72 and T4, T10, T24, or T48 (*p* < 0.01). Moreover, evaluating the different methods of conservations at the same time, we found a statistically significant worsening between the samples both for RNAlater and silica beads samples (*p* < 0.01) for T4, T10, T24, and T48, in comparison with controls.

Analyzing the samples maintained without any method of conservation, we found a significant influence of the temperature 72 h after the sampling (*p* < 0.01) for nuclei, inflammatory infiltrate, and for preservation of microscopic liver structure (*p* < 0.05).

### 3.2. DNA Analysis

DNA analysis allowed us to detect the expected 190 bp band for each sample, regardless of the selected preservation method, temperature, and time. No significant differences were observed among the samples.

### 3.3. RNA Analysis

RQI (RNA quality indicator) values were higher than 4, which reflect an acceptable quality of RNA at any time and any temperature for samples preserved with silica beads. A statistically significant positive effect of the preservation method (*p* < 0.0001 for 4 °C and 24 °C) on RNA integrity for silica bead samples was revealed. RNA quality was significantly higher in samples stored with silica beads at any time (Figure 3 and Figure 4) compared to samples stored without any method of conservation. Under-vacuum-preserved samples gave acceptable results only for samples stored at 4 °C and until 10 h after sampling (*p* < 0.05). After 10 h, RQI was <4 for all the samples at any temperature (Figure 3 and Figure 4).

## 4. Discussion

In terms of the samples stored without any method of conservation, obtained results showed histologically poorer scores at 72 h, in accordance with previous studies, enhancing the impact of time and temperature on the autolytic process [29].

RNAlater and silica bead-preserved specimens showed a worsening of histological scores at any temperature and after a very short time, already at 10 h. Silica beads caused sample shrinkage, already appreciable after 4 h, making this method unsuitable for histological analyses. Nevertheless, in some fields, such as paleopathology, mummified human or animal remains, after adequate rehydration, became suitable for light microscopy and immunohistochemistry analysis [30,31,32].

The efficiency of RNAlater as histological fixative is controversial. Results of this work strengthened the thesis that RNAlater is an unreliable method of conservation when histopathological evaluation is required [10].

Interestingly, under-vacuum packing showed acceptable histology results, especially at low temperature, in accordance with previous works, in which it was used as a reliable method to transport samples from surgery to pathology laboratories, at 4 °C [20,21].

Regarding the results from the biomolecular analysis, the expected DNA band of 190 bp was detectable for each sample analyzed. RNAlater, as expected, showed good RNA quality results, even at 24 °C after 72 h, accordingly to the manufacturers’ statements.

Results of RNA conservation in silica beads were worthy of interest. Indeed, there was a statistically significant positive effect of the preservation method (*p* < 0.0001 for 4 °C and 24 °C) on RNA integrity, in agreement with previous research [23]. In the light of these considerations, silica beads can be a more reliable, cheaper, and easier alternative to RNAlater for sampling when RNA analyses are requested.

RNA integrity was not well maintained in vacuum-sealed samples. Vacuum can be considered a suitable conservation method for RNA analysis only for a very short time, until 10 h after sampling, and refrigeration is mandatory for a good RNA quality. This method has been widely tested in the food industry, but not sufficiently studied for the preservation of tissues for research purposes, especially in terms of biomolecular analysis.

## 5. Conclusions

In conclusion, the selection of the right preservation method appears to be more important than the time of storage. The obtained results do not allow for the identification of an efficient and unique conservation method suitable for both histological and molecular investigations. However, the present findings should be taken into account to choose the right conservation methods according to what analysis should be performed. In fact, these methods can be synergistically used when multiple samples must be harvested to lower the cost and simplify the sampling, obtaining high-quality samples for both histological and molecular analyses (Figure 5). Further investigations should be made, evaluating the potential of silica beads in nucleic acid conservation for a longer time and at different temperatures, and other analysis should be made on different tissues. Rehydration of tissue for histological and immunohistochemical analysis should be considered.

## Figures and Tables

**Figure 1 animals-11-00649-f001:**
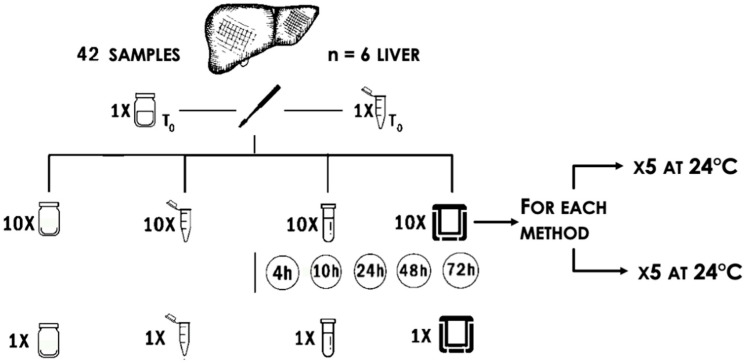
Schematic representation of the sampling method. Forty-two punch biopsies were collected. One sample was immediately fixed in 10% neutral buffered formalin and one in RNAlater as controls for histological and biochemical analysis, respectively. From left to right, 10 samples were stored without any preservation treatment, 10 in RNAlater, 10 in silica beads, and 10 under-vacuum. Half of the samples collected with each method were stored at 4 °C, half at 24 °C. After 4, 10, 24, 48, and 72 h, one sample from each group was split into two fragments, one was formalin-fixed, and the second was stored at −80 °C. This procedure was repeated for six livers.

**Figure 2 animals-11-00649-f002:**
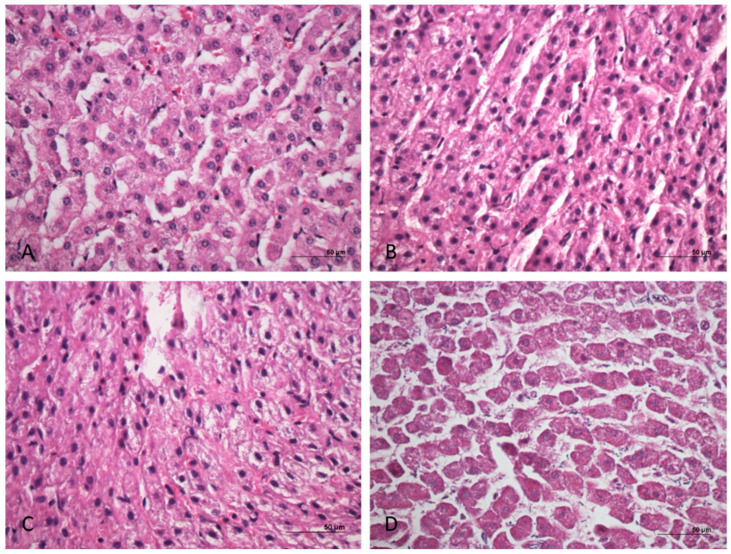
Cattle liver. Examples of histopathological scoring model. (**A**) Easily identifiable rounded nuclei, with one or more big nucleoli, multifaceted cytoplasm, and without alterations (nuclei and cytoplasm: score 5). (**B**) Mild hydropic degeneration and intracellular vacuoles. Nuclei in eccentric position (nuclei and cytoplasm: score 4). (**C**) Structural alteration of the liver. Indistinguishable nucleoli (nuclei and structure: score 3). (**D**) Pyknotic or absent nuclei and severe structural alteration (nuclei and structure: score 1). Hematoxylin and eosin (H&E).

**Figure 3 animals-11-00649-f003:**
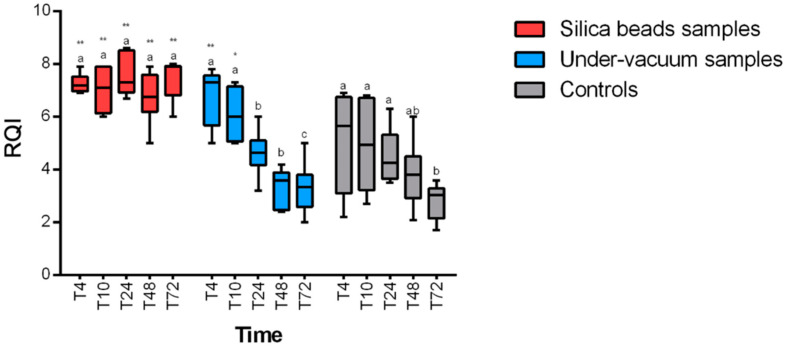
RQI (RNA quality indicator) results at 4 °C for the different preservation methods. Intra-group statistically significant differences from relative control (T0) are marked as follows: * *p* < 0.05, ** *p* < 0.01. Inter-group differences are marked with letters.

**Figure 4 animals-11-00649-f004:**
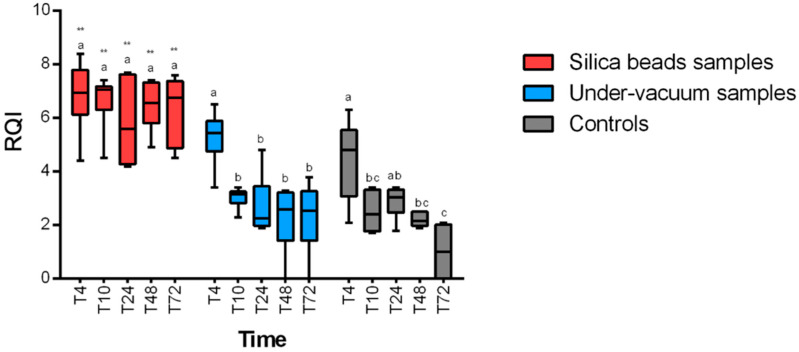
RQI (RNA quality indicator) results at 24 °C for the different preservation methods. Intra-group statistically significant differences from relative control (T0) are marked as follows: * *p* < 0.05, ** *p* < 0.01. Inter-group differences are marked with letters.

**Figure 5 animals-11-00649-f005:**
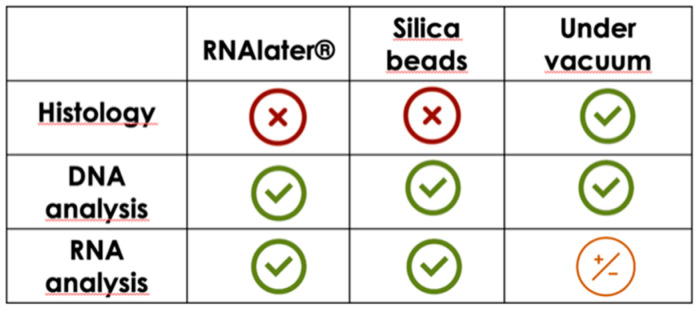
Schematic representation of the results in order to better select the right preservation method for histology or biomolecular analysis.

**Table 1 animals-11-00649-t001:** Histologic evaluation of microscopic slides: description of the scoring model used.

Score	Nuclei	Cytoplasm	Structure	Inflammatory Infiltrate
1	Absent	Marked loss of cellular integrity	Unrecognizable	Unrecognizable
2	Fading	Loss of cellular integrity	Hard to be recognized	Hard to be assessed
3	Shrinked	Vacuolized	Recognizable	Recognizable
4	Rounded, with coarsed chromatin	Multifaceted, with some vacuolization	Preserved	Easily recognizable
5	Rounded, with evident nucleoli	Multifaceted, without vacuolization	Perfectly preserved	Very easily recognizable

## Data Availability

No new data were created or analyzed in this study. Data sharing is not applicable to this article.

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
