# Peer review of "Coping with Tissue Sampling in Suboptimal Conditions: Comparison of Different Tissue Preservation Methods for Histological and Molecular Analysis"

_animals, 2021, doi:10.3390/ani11030649_

Round 1
Reviewer 1 Report
The aim of this study was the identification of an efficient, non-toxic, economic, easily portable and easy-to-use method for tissue preservation, suitable for “in field” sub-optimal conditions, capable to improve the performance of tissue sampling also in forensic medicine and reliable for both histological and molecular analysis.
It is an interesting and important work that can serve, I think, as a reference for future research. As strengths I point out the manuscript as a whole. In fact, is very well written, its sections are presented in a balanced, coherent and in an assertive way. The Material and Methods and Results are very well presented and supported in figures and tables. The data are properly discussed and compared with the results of other authors.
I am of the opinion this work justifies its publication, and which I recommend. I suggest only minor corrections before its publication, nothing conceptual or methodological. Very easy and pleasant work to read, and very useful for the future.
INTRODUCTION
Assertive and very well written properly framing the problem/study.
Comment 1 (lines 97-98): I only have a minor suggestion, and the authors can accept it or not: to absolutely justify the publication in Animals I suggest consider that these methods can also be applied in veterinary toxicology/poisoning, whether in field conditions, or others, as it may take some time between the cause of death/contamination and the collection of the tissues for later histopathological evaluation.
MATERIALS AND METHODS
Comment 2: Very well explained in such a way that they can be reproduced in other laboratories by other researchers.
Comment 3: In the text, the authors refer 5 criteria (line 136) but in the Table 1 only 4 of them appear, there is no (e) integrity and morphology of section borders. How was it evaluated?
RESULTS
Comment 4: Fig. 2 Photomicrography B is the least achieved and that the authors could try to improve, with another magnification in which the observed changes were more evident (does not have a scale).
Comment 5: Description appears for 4 of the indicated parameters (a), (b), (c), (d) and not for the (e) integrity and morphology of section borders, as already mentioned in the MM section (Table 1). The authors consider that it should be maintained (lines 136 and 139)?
CONCLUSIONS
Comment 6 (Line 304): Further investigations should include other tissues.
Reviewer 2 Report
Dear authors,
I considered manuscript as eligible for publication with a few minor corrections (listed below). It is of considerable value to present rather methodological results than an explorative approach, which improves everyday laboratory work.
This data, crucial on the first step of the experimental process -data collection- is equally essential and challenging to find/verify by other means than personal experience.
Although in comparison to other research articles presented manuscript lack of "novelty", it should be published for the benefit of us all.
I find a few aspects I kindly ask for clarification:
1] please put some additional effort to re-phrase some English sentences. First three chapters start from the same sentence, authors must avoid that.
2] Trizol extractions, in my opinion, should be at least briefly described. The manuscript should give essential information about its methods.
3] Figure 2: why there are differences in magnification? It makes it difficult to compare or understand the presented information. Picture B lacks a scale bar.
4] RQI data- why there is no data from Rna later in this publication? Authors describe its use in previous chapters, and somehow it is missing in one of the most critical aspects of its use.
Kind regards,
